# A Comprehensive Update on Retinal Vasculitis: Etiologies, Manifestations and Treatments

**DOI:** 10.3390/jcm11092525

**Published:** 2022-04-30

**Authors:** Aniruddha Agarwal, Anne Rübsam, Lynn zur Bonsen, Francesco Pichi, Piergiorgio Neri, Uwe Pleyer

**Affiliations:** 1The Eye Institute, Cleveland Clinic Abu Dhabi, Abu Dhabi P.O. Box 124140, United Arab Emirates; aniruddha9@gmail.com (A.A.); ilmiticopicchio@gmail.com (F.P.); nerip@clevelandclinicabudhabi.ae (P.N.); 2Cleveland Clinic Lerner College of Medicine, Cleveland, OH 44195, USA; 3Department of Ophthalmology, Charité—Universitätsmedizin Berlin, Corporate Member of Freie Universität Berlin and Humboldt-Universität zu Berlin, 13353 Berlin, Germany; anne.ruebsam@charite.de (A.R.); lynn.zur-bonsen@charite.de (L.z.B.); 4Department of Ophthalmology, Khalifa University, Abu Dhabi P.O. Box 124140, United Arab Emirates; 5Klinik für Augenheilkunde, Charité—Universitätsmedizin Berlin, Augustenburger Platz 1, 13353 Berlin, Germany

**Keywords:** retinal vasculitis, uveitis, ocular inflammation, fluorescein angiography, eye

## Abstract

Retinal vasculitis is characterized by inflammatory involvement of retinal arterioles, venules and/or capillaries and can be associated with a myriad of systemic and ophthalmic diseases. In this review, we have comprehensively discussed the etiologies, clinical manifestations, and presentations of retinal vasculitis. We have also included newer advances in imaging in retinal vasculitis such as OCTA and widefield imaging.

## 1. Introduction

Retinal vasculitis is characterized by inflammatory involvement of retinal arterioles, venules and/or capillaries and can be associated with a myriad of systemic and ophthalmic diseases [1,2,3,4,5]. The anatomy of the eye allows for direct visualization of the retinal vasculature through clinical examination, permitting evaluation of the structural alterations. The changes due to retinal vasculitis include sheathing, perivascular cuffing and exudation, vascular occlusion leading to capillary non-perfusion, pre-retinal hemorrhages, and neovascularization. If the disease continues to progress, it can lead to blinding complications such as tractional retinal detachments, vitreous hemorrhage, and glaucoma due to neovascularization of the iris/retina [1].

Since vascular supply to the retina is critical given its high metabolic activity, any compromise due to vasculitis can have permanent deleterious complications. Therefore, prompt assessment of the retinal vasculature, and evaluation using techniques such as fluorescein angiography (FA) and optical coherence tomography angiography (OCTA) is necessary [6]. The choroidal vasculature is assessed using indocyanine green angiography (ICGA). However, to date, there is still no generally accepted classification of retinal vasculitis. Retinal vasculitis has been classified based on etiology (infectious versus non-infectious), vascular caliber (large, medium, or small vessel disease) and clinical appearance (occlusive or non-occlusive) [1,2]. 

Since retinal vasculitis can be secondary to several systemic and ocular diseases, we have discussed the disease by classifying it based on the etiology (Table 1). In this review, we have comprehensively discussed the etiologies, manifestations, and presentations of retinal vasculitis (Figure 1). We have also included newer advances in imaging in retinal vasculitis such as OCTA and widefield imaging.

## 2. Infectious Retinal Vasculitis

### 2.1. Bacterial Retinal Vasculitis

Retinal vasculitis can be caused by several bacterial agents including tuberculosis, syphilis, Lyme’s disease, Brucellosis, Bartonella, and Leptospirosis. The rare causes of bacterial retinal vasculitis include Whipple’s disease. In addition, retinal vasculitis can also occur in bacterial exogenous or endogenous infections such as endophthalmitis and pan ophthalmitis [1,2,3,4,5,6,7]. 

#### 2.1.1. Tubercular Retinal Vasculitis 

Retinal vasculitis has long been recognized feature of intraocular tuberculosis (IOTB). This condition typically affects young Asian Indian males. The phlebitis is accompanied by vitritis, sheathing of the vessels, active/inactive choroiditis lesions, cystoid macular edema, capillary non-perfusion, and complications due to retinal neovascularization [8,9,10]. Occlusive retinal vasculitis with choroiditis is an important clinical manifestation of this disease. Figure 2 illustrates a patient with TB-related retinal vasculitis with fundus (Figure 2A) and FA findings (Figure 2B). 

There are numerous challenges in diagnosing IOTB and retinal vasculitis. There is a particular emphasis on the supportive laboratory tests including positive Mantoux (TST), chest radiograph findings, interferon gamma release assay (IGRA), and response to empiric ATT. Recently, there have been guidelines published to interpret these tests in the context of IOTB [9]. Since the condition is characterized by widespread occlusion of vessels, we often use widefield and ultra-widefield imaging (including FA) to diagnose the condition. These modalities can reveal findings such as peripheral capillary non-perfusion areas, retinal neovascularization, and retinal vascular leakage that can be easily missed. Treatment with scatter laser photocoagulation can be guided by these imaging tests. Tubercular retinal vasculitis may be accompanied by perivascular choroiditis, which is usually along the venous tracts [10,11]. 

Retinal vasculitis due to IOTB is usually treated with systemic corticosteroids and ATT. ATT with a multidrug regimen consisting of first-line anti-TB agents (isoniazid, rifampin, ethambutol, and pyrazinamide) is used. The second-line agents include levofloxacin, streptomycin, kanamycin, among others [8,9,12]. If the inflammation is mild and non-vision threatening, corticosteroids may be avoided altogether. In cases with severe retinal vasculitis, addition of systemic immunosuppressive agents may be considered. Complications such as retinal neovascularization can be treated with anti-vascular endothelial growth factor (anti-VEGF) injections and laser photocoagulation based on FA findings [13]. 

#### 2.1.2. Retinal Vasculitis Related to Syphilis

Syphilis is a multisystem venereal disease caused by *Treponema pallidum*. Ocular involvement in syphilis is common in secondary and tertiary stages. Recently, ocular syphilis has been considered to be a re-emerging disease. Approximately 4–5% of patients with secondary syphilis can develop ocular manifestations. The multisystemic involvement and variety of manifestations make the diagnosis of syphilis challenging. Ocular involvement can also be difficult to diagnosis, since the inflammation can affect both the segments of the eye, retinal vasculature, optic nerve, and the orbit (the great masquerader). Focal/multifocal chorioretinal lesions are common manifestations in the fundus. Necrotizing retinitis can occur, mimicking acute retinal necrosis. These present with either single or multiple yellowish-white patches of necrosis. There could be associated retinal vasculitis, vitritis and anterior chamber cells. Figure 3 shows FA of a representative case of ocular syphilis with retinal vasculitis and chorioretinal lesions. Retinal vascular ischemia can occur commonly (similar to IOTB). Severe inflammation results in frosted branch angiitis. 

The imaging findings of ocular syphilis are also protean. FA can show the presence of cystoid macular edema and optic nerve inflammation. Long-standing disease can show retinal photoreceptor disruption with subretinal fluid. Retinal vasculitis in syphilis commonly has associated macular edema. 

The preferred treatment of syphilis is penicillin G therapy (all stages of the disease). However, there may be variations in the duration of the therapy based on the stage. Secondary and tertiary syphilis require longer therapy with antibiotics. The recommended dosing schedule for ocular syphilis is either aqueous penicillin G or procaine penicillin G (both for 10–14 days). Probenecid (500 mg four times/day) can be added for ocular disease. In several countries, availability of penicillin G could be a challenge. In addition, allergies to penicillin are common. Therefore, the alternate agent used in ocular syphilis is ceftriaxone (2 g/day intravenous for 2 weeks), which shows good results.

#### 2.1.3. Retinal Vasculitis Associated with Lyme’s Disease

Lyme disease is a bacterial disease caused by a spirochete, *Borrelia Burgdorferi.* The transmission of this bacteria is facilitated by the ticks of *Ixodes ricinus*. Lyme disease is prevalent in the United States and Europe, and certain countries in Asia. There is a geographic distribution of Lyme disease (potentially related to the presence of the arthopod vector). In the US, more than 90% cases are reported from Connecticut and Massachusetts [14,15,16]. 

The ocular manifestations of Lyme disease can happen any time during the course of the infection (from days to several years). After the primary infection, the patients can present with ocular inflammation (most commonly, intermediate uveitis or posterior uveitis). Retinal vasculitis is a common association with these forms of uveitis. Retinal vasculitis may be accompanied by multifocal choroiditis. Unlike IOTB, the choroiditis lesions of Lyme disease are small, punched-out with variable grades of vitreous haze. FA is useful in detecting active retinal vasculitis characterized by retinal vascular leakage. Unlike IOTB and syphilis, occlusive retinal vasculitis is rare in Lyme disease. Cystoid macular edema is present in the majority of patients and can be diagnosed on FA and OCT [15,17,18,19,20].

Broad-spectrum antibiotics such as beta lactams and tetracyclines are used for the treatment of Lyme disease. For the ocular inflammation, corticosteroids are added under antibiotic cover. Patients with mild disease and erythema migrans can be treated with oral doxycycline (100 mg twice/day) or other agents such as amoxicillin (500 mg three times a day), or cefuroxime (500 mg twice/day) for 2 weeks. Lyme disease in children is treated with doxycycline (4–8 mg/kg/day), or amoxicillin (50 mg/kg/day). Local corticosteroids can also be used to treat retinal vasculitis [19,20].

#### 2.1.4. Bartonella-Related Retinal Vasculitis

Bartonellosis (cat-scratch disease—CSD) is caused by a Gram-negative bacillus (*Bartonella Henselae*). CSD is also spread by Ctenocephalides felis (cat flea, an arthropod vector) through domestic cats that act as the reservoir. Infection to humans is transmitted by scratch from a cat or contamination of surface wounds [21,22,23].

CSD-related uveitis presents as retinochoroidal inflammation and associated retinal vasculitis. The inflammation of the retinochoroid presents as juxtapapillary chorioretinitis. One characteristic finding of retinochoroidal lesions in CSD is the presence of angiomatous lesions that represent abnormalities involving the retinal vasculature. Such similar changes are also observed in other integumentary systems such as nail beds in CSD. Abnormalities such as bacillary angiomatosis, branch retinal arteriole occlusion (BRAO), or retinal vasculitis can be diagnosed in these patients. Figure 4 shows fundus photographs (Figure 4A,B), and OCT (Figure 4C,D) of a patient with CSD-related BRAO. Occlusive retinal vasculitis is also a known feature of CSD on FA [21,22,23].

CSD is usually treated with systemic antibiotics. Broad-spectrum antibiotics include azithromycin, trimethoprim–sulfamethoxazole, rifampin, and others that are usually employed. Patients with ocular inflammation may be administered concomitant corticosteroids (topical or oral prednisone). The corticosteroids are usually given only under the cover of antibiotics mentioned above. Other antibiotics employed in CSD include ciprofloxacin and gentamicin. Azithromycin can be given in patients with moderate to severe disease [21,22,24,25].

#### 2.1.5. Miscellaneous Bacterial Retinal Vasculitis

Whipple’s disease is caused by the bacterium, *Tropheryma whipplei*, which is a weakly Gram-positive bacterium that causes ocular manifestations such as posterior segment disease and rarely infective endocarditis. Due to its rarity and non-specific clinical features, Whipple’s disease is difficult to diagnose clinically. Therefore, this condition may be undiagnosed for several years [26]. In order to achieve the correct diagnosis, most patients require invasive retinal biopsies and special stains to diagnose the bacteria. The most common ocular manifestations of Whipple’s disease include diffuse retinal vasculitis and retinochoroidal lesions. The retinal vasculitis may be associated with hemorrhages, sheathing, retinal capillary occlusion, capillary dilation at the optic disc, and choroidal folds (Figure 5) [26,27,28]. 

### 2.2. Viral Retinal Vasculitis

#### 2.2.1. Herpes-Family Related Retinal Vasculitis

Retinal vasculitis can be caused by several viruses from herpes, including herpes simplex virus (HSV), varicella zoster virus (VZV), cytomegalovirus (CMV), and Epstein–Barr virus (EBV). The manifestations of retinal infection by these viruses can have varied manifestations, but there may be an overlap of clinical presentations. Retinal vasculitis is present with retinitis or retinal necrosis [29]. 

The ocular manifestations of viral retinitis such as retinitis can rapidly progress and result in complete blindness due to retinal necrosis and other complications such as rhegmatogenous retinal detachment. Necrotizing retinitis, known as acute retinal necrosis (ARN), is the most common and feared manifestation of viral posterior segment disease (Figure 6 shows an ultra-widefield image of a patient with ARN). Rhegmatogenous retinal detachments are common once the retinitis lesions have caused retinal atrophy and are difficult to manage even surgically. While viral retinitis is common in subjects who are immunocompromised due to conditions such as HIV infection and acquired immune deficiency syndrome (AIDS), these infections can occur among patients with solid organ transplants or other systemic diseases. The other presentations of viral retinitis include CMV retinitis and progressive outer retinal necrosis (PORN). Subjects with other conditions such as lymphocytic proliferation and loss of cutaneous anergy are also at risk of developing retinitis and its related complications. There are cases of immunocompetent individuals also developing viral retinitis in the literature [29]. 

The characteristic features of ARN include severe, vision-threatening ocular disease with rapid evolution, retinal necrosis, and loss of the neural retina. The presence of occlusive retinal arteritis is a hallmark of the disease and an important clinical feature (Figure 6). There could be anterior chamber cells and vitritis depending on the systemic immune status. Viral retinitis lesions are usually peripheral in location. The retinitis lesion is typically described as a focal area of full-thickness necrotic lesion with extensions and occlusive vasculitis with arteriolar narrowing and retinal hemorrhages. The active lesions are known to rapidly progress circumferentially toward the posterior pole and central macula [30,31,32,33]. 

CMV retinitis occurs usually as an opportunistic infection in patients with immunocompromised states such as HIV/AIDS (most commonly), or in patients with other comorbidities mentioned in the preceding paragraphs. CMV retinitis lesion is a full-thickness necrotizing retinitis (granular in appearance compared to ARN lesions) in the peripheral retina. These lesions tend to expand centrifugally. The retinitis is accompanied by hemorrhage and vasculitis. Due to its appearance and presence of hemorrhages, the lesions of CMV are termed as pizza pie retinopathy. CMV retinitis can also present with an exuberant vasculitis (known as frosted-branch angiitis) [29,34,35,36,37,38].

ARN needs prompt treatment based on clinical assessment with intravenous acyclovir 10 mg/kg every 8 h/day. This is followed by a maintenance dose of oral acyclovir 800 mg 5 times every day. Recently, studies have shown similar efficacy of oral treatment with antiviral therapy in ARN. Agents such as valacyclovir and famciclovir can be given orally. Either oral valacyclovir 1 gm three times a day or famciclovir 500 mg 3 times daily can be successfully used in the treatment of ARN without intravenous therapy [29,31,32]. 

CMV retinitis is treated based on the cause of immunocompromised status. In patients with HIV/AIDS and low CD4 counts, therapy with highly active antiretroviral therapy (HAART) is important in improving the overall immune status of the patients. Not only CMV retinitis, but HAART, is beneficial for improving the outcomes of other opportunistic infections. Intravitreal injections of ganciclovir (4 mg/0.1 mL) are used as an adjunctive therapy in patients with macular or vision threatening disease. Anti-CMV therapy may be discontinued if the CD4 count improves to ≥100 cells/µL [34,35,37].

#### 2.2.2. Dengue, West Nile Fever, and Other RNA Viruses

RNA viruses form a large family of infectious agents such as dengue fever virus, West Nile fever (WNV) viruses and Chikungunya virus. The infections caused by these agents can be associated with retinal vasculitis. RNA viruses (most importantly dengue fever, Chikungunya, influenza, Zika virus) have been recognized as re-emerging diseases in several countries [39]. Dengue fever can cause several uveitic manifestations including retinal whitening/edema, hemorrhages, and retinal perivasculitis. Vision loss in dengue retinopathy could be due to macular edema, pre-retinal/intraretinal hemorrhages, dengue-related foveolitis, and microvascular changes. Rare manifestations of dengue retinopathy include acute macular neuroretinopathy (AMN) [40,41,42,43,44]. 

West Nile virus is an RNA arbovirus belonging to the Flaviviridae family transmitted from infected birds to humans by mosquitoes causing WNF. Invasive WNV include meningitis, encephalitis, acute flaccid paralysis. WNV ocular manifestations include bilateral classical multifocal nummular creamy chorioretinal curvilinear lesions with mild vitritis. Occlusive retinal vasculitis, optic neuritis, uveitis without focal lesions, and congenital chorioretinal scarring are other manifestations. Most cases with occlusive vasculitis have been associated with irreversible vision loss. Treatment ranges from observation for self-limiting course to intravitreal anti-VEGF for persistent macular edema [41].

Chikungunya virus can also lead to pyrexia and systemic involvement, along with ocular inflammation. Posterior segment includes mild vitritis, focal or multifocal retinitis, associated edema (subfoveal or intra-retinal fluid) and whitening. Other manifestations of severe disease include exudative retinal detachment, retinal vasculitis, intraretinal hemorrhage, optic neuritis, and rarely macular choroiditis [41,45,46]. 

#### 2.2.3. Human Immunodeficiency Virus

HIV-related retinal disease is an important cause of ocular morbidity worldwide despite the widespread availability of HAART. Apart from opportunistic infections, patients with HIV/AIDS can develop ocular manifestations directly due to HIV such as retinal microangiopathy. Microvascular changes due to HIV can lead to the development of retinopathy and focal ischemia. These changes result in development of cotton–wool spots, retinal hemorrhages and microaneurysms. Newer imaging modalities such as OCTA help in detecting retinal ischemia, which may otherwise go undetected clinically or on FA [37,47,48]. 

### 2.3. Fungal Retinal Vasculitis

#### 2.3.1. Candidiasis

Fungal infections such as endophthalmitis due to candida can occur either from an exogenous route following trauma (open globe injury) or surgery. Endogenous candida endophthalmitis from hematogenous spread is a more common presentation of this fungus. High-risk patients for endogenous infection include those with intravenous drug abuse, hyperalimentation, long hospital stay, COVID-19 infection related morbidity, diabetes, bone marrow transplantation, and malignancy, among others. Prolonged corticosteroid therapy is another important risk factor for candida endophthalmitis. 

Retinochoroiditis is a common feature of candida endophthalmitis and is characterized by single/multiple white retinal lesions in the superficial part along with vitreous inflammation and detaching hyaloid. The lesions are accompanied by vitreous cotton ball opacities. Retinal vasculitis is associated with candida retinochoroiditis. It can present with sheathing, cotton–wool spots, and superficial hemorrhages [49,50,51]. 

#### 2.3.2. Coccidioidomycosis

Coccidioidomycosis is an infection caused by dimorphic fungus Coccidioides immitis or Coccidioides posadasii. Coccidioidomycosis is a granulomatous disease and can involve multiple organ systems. Disseminated disease can result in ocular manifestations including uveitis, which is considered rare. The presentation usually consists of either idiopathic iridocyclitis or choroiditis in a person who has lived or traveled through endemic areas, including the USA (Arizona, Texas, parts of Central America), Argentina, Mexico, and California. Thus, the disease has a geographic distribution [52,53].

Coccidioidomycosis can present with juxtapapillary chorioretinitis with hemorrhages, exudates, and retinal vasculitis. The manifestations in the posterior segment are varied, and some patients can develop perivascular punched out lesions along the vessels [52,53]. 

### 2.4. Parasitic Retinal Vasculitis

#### 2.4.1. Toxoplasma-Related Retinal Vasculitis

Toxoplasmosis is a systemic disease caused by a parasite *Toxoplasma gondii.* Cats serve as a definite host for this infection. Toxoplasmosis has a high prevalence rate; in the US, it is estimated that 22.5% of the population more than 12 years age are already infected. Among these, 10% individuals can present with ocular toxoplasmosis (OT) [54,55,56].

OT can occur because of congenital or acquired infection. Congenital disease can have a recurrence later in life, resulting in active OT necessitating treatment. OT due to congenital infections can present early with retinochoroidal healed scars (active retinal disease is rare). The manifestations of OT include retinochoroiditis (Figure 7), retinal vasculitis, occlusive retinal vasculitis, papillitis, and neuroretinitis. The lesions of retinochoroiditis due to OT appear as foci of retinitis adjacent to a chorioretinal scar. The lesions are accompanied by dense vitritis overlying the lesion (termed as *headlight in fog appearance*). Retinal vasculitis may be associated with the OT lesion. Venous involvement is more frequent than arterial disease in OT. Severe retinal vasculitis can be observed in areas of the fundus away from the site of the retinochoroiditis. Inflammatory vascular occlusions are common in eyes with OT [54,55,56,57,58,59,60]. 

FA is useful for the assessment of lesions of OT. Early FA shows hypofluorescence in the area of the active lesion. In the late phase, there is hyperfluorescence from the periphery of the lesion to its center. Active vasculitis presents with intense leakage and cystoid macular edema [55,56,58,60].

OT is typically treated with systemic antibiotics given for 4 to 8 weeks. Classic therapy of OT consists of pyrimethamine (25 mg–50 mg daily orally) and sulfadiazine along with corticosteroids. Folinic acid is also added to the regimen. Trimethoprim–sulfamethoxazole (160 mg–800 mg twice daily) is an alternate option in patients who cannot tolerate pyrimethamine. Trimethoprim–sulfamethoxazole also has advantages of low cost and wide availability. The combination of trimethoprim–sulfamethoxazole may also prevent recurrences of OT. Clindamycin (300 mg orally four times daily) can be added to the triple therapy (making it quadruple therapy). Antimalarial drugs including artimiside and artemisone may be employed in the treatment of OT [56,59,60,61].

The intravitreal therapy for toxoplasma retinochoroiditis consists of a combination of intravitreal clindamycin (1 mg/0.1 mL) with dexamethasone. This therapy can be given in all cases and is comparable with systemic clindamycin, or other therapies for toxoplasmosis, although clinical trials comparing efficacy have not yet been performed. Intravitreal injections can be given bi-weekly or every week until resolution of the lesions [61,62]. 

#### 2.4.2. Miscellaneous Parasitic Retinal Vasculitis

Several parasitic infections of the eye can result in retinal vasculitis in addition to other retinochoroidal abnormalities. Toxocariasis is associated with granulomas in the posterior segment. These granulomas can have associated retinal vasculitis. In addition, amoebiasis can also result in retinal vasculitis. The treatment for these conditions generally is for the underlying disease, and no specific therapy is needed for retinal vasculitis [1,2]. 

## 3. Non-Infectious Retinal Vasculitis 

### 3.1. Retinal Vasculitis with Systemic Involvement

#### 3.1.1. Behcet’s Syndrome

Behcet’s syndrome (BS) (also known as Admantiades Behçet’s syndrome or Behçet’s disease) is an uncommon systemic vasculitis that often bears a high risk of serious complications. According to the Chapel Hill nomenclature, BS is classified as a disorder affecting variable vascular calibers. Arteries can be affected, whereas venous involvement predominates [63]. The systemic disease is characterized by oral aphthae (Figure 8), genital ulcers, thromboembolic manifestations at various sites and inflammatory ocular changes. A life-threatening course can develop with involvement of, e.g., the airways or central nervous system. Accordingly, early diagnosis and adequate anti-inflammatory treatment are required. 

Ocular involvement is often the first and dominating clinical finding. Retinal vasculitis, which affects both the venous and arterial blood supply, is a hallmark of BS. Venous branch occlusions with intraretinal bleeding and macular edema are common and threaten central visual acuity. They were observed in up to one third of patients with ocular BS. In contrast, central vein occlusion (about 4–5%) and arterial occlusions are less common [64]. Vascular ischemia is observed as a rare (about >1%) but prognostically unfavorable sign (Figure 9 shows a fundus photograph of a representative case with severe retinal atrophy following occlusive retinal vasculitis) [65]. Not infrequently, this leads to neovascularization due to ischemia and must be considered accordingly (laser coagulation) in addition to anti-inflammatory therapy.

Results dating back to the 1980s show that up to 70% of affected eyes suffer from a significant reduction in visual acuity (<0.1) within 5–10 years [66]. When left untreated, this resulted often in blindness in young men. New treatment options, especially those such as biologics, were able to significantly improve the prognosis [67,68,69,70]. In general, therapy of BS today includes a wide spectrum of anti-inflammatory and immunomodulatory agents with expanding importance of biologics and “small molecules” [68,71]. In 2018, the update on the EULAR therapy recommendations for BS was published [72]. Ultimately, BS is treated in a differentiated manner adapted to the severity of the existing findings in an individual patient.

Clinical studies show that the concentrations of TNF and the soluble TNF receptor in serum and aqueous humor are increased in BS patients. Accordingly, adalimumab and infliximab have been introduced and successfully used. Since adalimumab has received EMA/FDA approval for the treatment of intraocular inflammation of the posterior eye segment, this agent is often preferred today.

Treatment with interferon α for ocular BS has also been proven already for some time and has been reviewed in 2004, proposing a good response in BS uveitis (about 90% remission). Noteworthy was the favorable effect on cystoid macular edema, which was recently found superior to cyclosporin and GC in an RCT trial [73]. In severe cases with CNS involvement, cyclophosphamide had previously been recommended, but TNF antibodies (infliximab) are now considered as first line therapy, possibly followed by maintenance therapy cDMARDs, e.g., methotrexate or azathioprine [68,74]. For vascular complications, e.g., presenting as recurrent thrombosis, anticoagulation is recommended but only as a passage treatment (for 6 months), together with immunosuppressive therapy. In addition to dysfunction of the vascular endothelium, changes in the coagulation and fibrinolytic systems were observed. Antibodies targeting endothelial cells were detected in up to half of BS patients and could be correlated with disease activity. Elevated levels of VEGF and MCP-1 were seen in BS patients and thrombosis [75]. After control of the inflammatory activity, there is no longer an increased risk of thrombosis, unless additional risk factors such as cardiolipin antibodies or other thrombophilic factors (factor V Leiden mutation or similar) are present. 

In principle, locally applied therapies are effective in ocular BS. Intravitreal injections of depot preparations and inserts, e.g., dexamethasone or fluocinolone into the vitreous cavity, can be an option during persistent macula edema, concomitant infections (e.g., TB) or pregnancy, which may limit the use of immunosuppressive therapy.

#### 3.1.2. Systemic Lupus Erythematosus

Systemic lupus erythematosus (SLE) is considered a secondary vasculitis that can affect different vascular calibers. Systemic findings in SLE include arthritis, myocarditis, nephritis fever, and Raynaud’s phenomenon. As important risk factors indicating a more problematic clinical course, involvement of the central nervous system, eye and kidney are known. In addition, the occurrence of antiphospholipid antibodies may result in more severe manifestations. 

Eye involvement in SLE may also affect the outer eye, e.g., the conjunctiva; but retinopathy remains the feared finding. The incidence of retinopathy is reported to affect up to 30% of patients [76]. Retinal involvement may be the first symptom of the disease [77]. and therefore is an important differential diagnosis at initial presentation. Retinopathy primarily affects the posterior pool with cotton–wool spots, frosted branch angiitis, retinal bleeding and vascular fibrosis. In addition, optic nerve atrophy due to vasculitis may occur and impress as optic neuritis [78]. Secondary changes are often limiting visual acuity, e.g., vitreous bleeding seen in up to 40% of SLE retinopathy patients with severe courses [79], as well as varicella central vein occlusions [80]. Larger retinal vessels can also be affected, which can lead to retinal central arterial occlusion, a complication that is otherwise rarely seen in retinal vasculitis [81]. Remarkably, commonly, there are no intraocular signs of inflammation presenting as uveitis such as vitreous haze or vitreous cells.

The exact pathogenesis of SLE is unclear. Involvement of immune complexes and complement activation as well as fibrinoid degeneration of the vessels are suspected [82]. Therefore, no specific treatment is available. Treatment of SLE includes immunosuppression and anticoagulation. The latter is particularly indicated in patients with antiphospholipid antibodies. Plasmapheresis was also recommended in therapy refractory cases [83]. In patients with occlusive vasculitis and NV, laser coagulation is indicated.

#### 3.1.3. Uveitis Associated with Multiple Sclerosis

Multiple sclerosis (MS) is a chronic autoimmune inflammatory disease of the central nervous system (CNS) consisting of inflammation, demyelination, and neuroaxonal degeneration along with gliosis [84]. It predominantly affects females in the 20–40 years of age with an estimated female-to-male ratio of 3.5:1 [85]. MS is diagnosed using the revised McDonald diagnostic criteria from 2017, combining clinical, radiological and laboratory evidence of dissemination in time and space and a proof of positive assay for cerebrospinal fluid (CSF) oligoclonal bands (OCBs) [86].

Ocular manifestations of MS include optic neuritis, motor abnormalities, internuclear ophthalmoplegia, nystagmus, and uveitis [87,88,89]. In large epidemiological studies, the frequency of uveitis among patients with MS ranges from 0.65% to 1.1% [87]. However, MS is reported among patients with uveitis, with a frequency of 0.9% to 1.7% [87].

MS and uveitis are both associated with expression of HLA-DRB1*15:01 haplotype, a subtype of HLA-DR2, suggesting underlying genetic factors [90]. In patients with MS-related uveitis, intermediate uveitis (IU) is the most frequent disease phenotype followed by panuveitis [91]. Peripheral vascular involvement is a consistent feature of idiopathic pars planitis and of MS-associated IU [92]. It is typically asymptomatic and best detected on ultrawide-field FA. The most frequent complications in uveitis associated with MS include cataract, CME, epiretinal membrane (ERM), glaucoma, and occlusive vasculitis. Figure 10 demonstrates a typical clinical presentation of MS-associated uveitis. In 2021, new classification criteria for MS-associated IU according to the SUN working group were published [93]. They combine the evidence of IU by (a) vitreous cells and/or vitreous haze, (b) less anterior chamber inflammation compared to vitreous if present, (c) no evidence of retinitis or choroiditis AND the evidence of MS using the updated McDonald diagnostic criteria. Furthermore, syphilis, Lyme disease and sarcoidosis should be ruled out. 

Uveitis can be the first presenting feature of MS in 36% patients [94]. Magnetic resonance imaging (MRI) of the CNS can be performed for highly suspected cases to have CNS demyelinating lesions, including patients with neurological symptoms suggestive of MS and presence of vasculitis on FA with or without granulomatous anterior segment inflammation. 

Similar to noninfectious IU, MS-associated uveitis is usually treated with systemic corticosteroids. However, data on the effects of currently available disease modifying therapies for MS on MS-associated uveitis are scarce. Retrospective studies have reported improvement of MS-associated uveitis following treatment with interferon-beta or glatiramer acetate [95], as well as mycophenolate mofetil [96]. Furthermore, case reports suggested improvement of refractory MS-associated uveitis following treatment with natalizumab [97], alemtuzumab [98], and the interleukin-1-receptor antagonist anakinra [99]. Despite the minor role for B-cells in MS, B-cell depleting anti-CD20 therapies such as rituximab and ocrelizumab have proven highly effective in the treatment of MS [100]. Of note, anti-TNF monoclonal antibodies and agents such as soluble TNF-alpha receptors have previously been associated with de novo demyelination and worsening of pre-existing demyelinating disease and are as such not recommended in patients with IU suspicious of MS [101].

#### 3.1.4. Susac’s Syndrome

Susac’s syndrome (SS) is a rare condition characterized by a classic triad of sensorineural hearing loss (SNHL), encephalopathy, and retinal branch retinal artery occlusions (BRAO). SS is commonly seen in young women, with a female/male ratio of 3:1 [102]. The diagnosis of SS is mostly delayed and overlooked because all components of the clinical triad are seen in less than 15% of patients at initial presentation [103].

The pathogenesis remains elusive. SS can be caused by an autoimmune inflammatory endotheliopathy that causes microangiopathy leading to vascular occlusion and ischemia affecting the precapillary retinal arterioles, the inner ear, and the brain [104]. Anti-endothelial cell antibodies (AECAs) are found in up to 30% of patients [105]. However, these AECAs are not specific to SS and can be found in other inflammatory diseases as well.

The most common initial neurological manifestation of encephalopathy becomes symptomatic as headache in 80% of cases [106]. Other symptoms are cognitive decline with loss of memory, concentration, disorientation, behavioral changes and less common seizures [106]. Brain involvement is demonstrated by typical MRI findings in T2 or fluid-attenuated inversion recovery (FLAIR)-weighted such as hyperintense, multifocal, round small lesions in the corpus callosum, referred to as “snowballs”, or “icicle” and “spoke”-like lesions [105,106].

The vestibulocochlear system involvement initially demonstrates as low frequency SNHL in 86% of patients, seen on audiogram [106]. With disease progression, high frequency SNHL ensues [107]. Other symptoms such as vertigo, nystagmus and tinnitus are often associated [106].

Ophthalmic findings in SS include focal retinal whitening/pallor, retinal vascular segmentation, arteriolar narrowing, cotton–wool spots, optic nerve head hyperemia, optic disc pallor, perivascular sheathing, peripheral capillary non-perfusion and arterial silver wiring [108]. Yellowish arterial wall plaques (Gass plaques) at mid-arteriolar segments are commonly found [109]. FA is mandatory in the diagnosis. The characteristic features on FA, which are present in 99% of patients, are recurrent multiple BRAO in the absence of retinal vascular inflammation, and segmental arterial wall hyperfluorescence [109]. OCT findings are a hyperreflective band in the inner retina in the area of the acute occlusion or retinal thinning consistent with retinal atrophy in previous artery occlusions. Figure 11 shows a case of SS with neuroimaging, FA and OCT findings. On OCTA, different degrees of capillary ischemia have been reported in superficial and deep capillary plexus, which clinically demonstrate as areas with visual field defects on perimetry [110].

Treatment of SS depends on the severity of CNS involvement [105]. First-line therapy of SS consists of high-dose intravenous corticosteroids followed by oral corticosteroids with slow tapering [105]. Treatment is usually combined with intravenous immunoglobulin, cyclophosphamide, mycophenolate mofetil, tacrolimus or rituximab. Treatment usually lasts for a long time (more than 2 years) for all patients regardless of severity category [105].

#### 3.1.5. ANCA-Associated Vasculitis

Vasculitides that are associated with antineutrophil cytoplasmic antibody (ANCA) formation (AAV) can be differentiated into three separate entities: granulomatosis with polyangiitis (GPA; formerly Wegener’s granulomatosis), microscopic polyangiitis (MPA), and eosinophilic granulomatosis with polyangiitis (EGPA; previously identified as Churg-Strauss syndrome).

The pathogenesis of ANCA-associated vasculitis is not understood completely. The pathogenesis involves both humoral and cell-mediated immune responses. Recent studies suggest that genetic background, risk of disease relapse, prognosis, and co-morbidities are closely linked to the ANCA serotype, e.g., proteinase 3 (PR3)-ANCA (or c-ANCA) and myeloperoxidase (MPO)-ANCA (or p-ANCA), associated with the disease phenotypes GPA or MPA. In particular, PR3-ANCA is related with GPA, while MPO-ANCA is predominantly associated in patients with MPA; however, there is also overlap in these disorders [111].

In 2012, the definitions for AAV were revised at the Chapel Hill Consensus Conference following their first declaration in 1994 (Figure 12) [112]. Granulomatosis with polyangiitis, MPA and EGPA have estimated annual incidence rates of 2–14, 2–10 and 0.5–4 per million, respectively [113]. They are commonly diagnosed in the elderly (>60 years) and are more common in men.

The spectrum of the AAV varies widely, and therefore, the clinical features can be different. Common clinical presentations of GPA include nasal involvement, epistaxis, intraocular inflammation, upper airway involvement and often kidney involvement. Patients with MPA are often older and often have more severe renal disease as compared to GPA and may present with rash and neuropathy. Eosinophilic granulomatosis with polyangiitis often appears as a multisystem disorder dominated by asthma, nasal polyposis, and blood eosinophilia. Since the disorder remains rare and because of its unspecific clinical presentation, AAV often remains a diagnostic challenge. Consequently, this often results in a delayed diagnostic of more than six months in approximately 30% of patients.

Ocular involvement has been reported in 50–60% of individuals with ANCA-positive vasculitis. Of note, in 8–16% of those affected, it has been the first clinical manifestation [114]. Eye involvement in patients with GPA is dominated by scleritis (75.0%) with non-necrotizing anterior scleritis as the most frequently reported subtype, followed by uveitis (18%) and other ocular inflammatory findings (34%), which include peripheral ulcerative keratitis, orbital cellulitis, dacryoadenitis and dacryocystitis [115]. Whereas retinal vasculitis might be assumed as a typical finding related to the systemic and ocular involvement, most GPA patients are affected by anterior uveitis. If the retina is affected, vascular findings may vary from benign cotton–wool spots to more severe vaso-occlusive disease. This includes retinal artery occlusion as well as ocular thrombosis [116]. In EGPA and MPA, ocular manifestations are rare and may present as orbital cellulitis in EGPA. In MPA, ocular involvement is more commonly seen as peripheral ulcerative keratitis or as vaso-occlusive retinitis [116].

Currently, systemic corticosteroids combined with cyclophosphamide or rituximab are the gold standard to initially treat these patients with severe, life-threatening disease [117]. Other first line treatments are based on methotrexate or mycophenolate mofetil. However, these approaches should be considered for patients only without any organ or life-threatening involvement [117].

#### 3.1.6. Sarcoidosis

Sarcoidosis presents as a systemic disease characterized by noncaseating granulomas. There is wide variation in terms of ocular manifestation and organ involvement, which may be among the reasons for the often postponed diagnosis. Ocular involvement has been reported between 6% and 79% in sarcoidosis, presenting as uveitis as the main eye manifestation [118]. Notably, the diagnosis of ocular sarcoidosis is determined by the revised International Workshop on Ocular Sarcoidosis (IWOS) criteria. This includes three groups—definite, probable, or presumed—according to the fulfillment of criteria as intraocular signs and systemic investigations [119,120,121].

Eye involvement usually presents with granulomatous changes, but vascular involvement is also often recognized as venous vascular sheading. A multifocal periphlebitis is seen in up to 37% of patients with ocular sarcoidosis presenting with segmental vascular shedding “candle wax dripping phenomenon” with exudation, which are seen in the peripheral retina (Figure 13) [122,123]. Retinal vasculitis is also one of the findings that was accepted as a disease at the ocular sarcoidosis workshop [119]. Although vasculitis in ocular sarcoidosis is usually not obstructive, it can also be seen in isolated cases with ischemic changes [124]. In addition, macroaneurysms, vascular occlusions and secondary neovascularization have all been observed. Vasculopathy in ocular sarcoidosis is one of the few identifiable associations in the context of intermediate uveitis [125]. Patients frequently present with inflammation in the vitreous cavity and vitreous infiltration described as “snowballs”. Similar changes are also seen in intermediate uveitis in the context of multiple sclerosis. Here also, peripheral peri phlebitis is a hallmark on fluorescein angiography in up to 76% of patients with concurrent intermediate uveitis [123].

Therapeutically, corticosteroids play a decisive role in both the short term and in basic therapy. They are used both as topical and oral applications, especially for anterior manifestations. Moreover, intraocular corticosteroid implants are an option to treat chronic macular oedema or be considered in the case of unilateral findings. Disease-modifying anti-rheumatic drugs (DMARD) and biologics are used for long-term immunosuppression in relapses, especially for uveitis intermedia, posterior and panuveitis [126]. The most commonly used DMARD is methotrexate, followed by azathioprine, mycophenolate mofetil and cyclosporine. The TNF-alpha inhibitor adalimumab predominates in biological therapy [127].

### 3.2. Isolated Ocular Disorders (Vasculitis Limited to Ocular Manifestations)

#### 3.2.1. Idiopathic Retinal Vasculitis

Vasculitis limited to the eye can in turn be caused by infectious or non-infectious causes. In addition, it must be considered that some systemic vasculitis types can also primarily manifest in the eye and can appear as isolated ocular vasculitis (e.g., sarcoid). About 5–10% of isolated retinal vasculitides are considered idiopathic [4].

#### 3.2.2. IRVAN Syndrome

This also includes patients with IRVAN syndrome, presenting with typical changes. The diagnosis of IRVAN—idiopathic retinitis, vasculitis, aneurysms, and neuroretinitis—is based on a triad of clinical findings. These include three major criterium: retinal vasculitis, neuroretinitis and aneurysmal dilations at arterial bifurcations; and three minor criterium: macular exudation, capillary nonperfusion and neovascularization. Macroaneurysms can be considered as a disease-defining criterium for IRVAN syndrome and are a sine qua non condition for its diagnosis [128,129].

While IRVAN is understood as an isolated eye disease with a strict phenotype, systemic changes were also found in some of the patients, especially including the nervous system. Multiple sclerosis and ischemic strokes can be cited here [128]. Angiographic representations are necessary for diagnosis, and the wide-field FAG examination should be emphasized.

Based on current observations, early pan-retinal laser photocoagulation should be considered when angiographic evidence of widespread retinal nonperfusion is present; in addition, anti-VEGF is an efficient treatment option that significantly improves the prognosis.

#### 3.2.3. Birdshot Chorioretinopathy (BSCR)

In contrast to IRVAN, BSCR is well characterized [130]. It can be cited as a typical example of isolated, non-infectious, retinal vasculitis. Information on its incidence varies, as the disease mainly occurs in Caucasian patients; in the USA and Europe, a prevalence of about 5–9% is assumed. The bilateral, partly asymmetrically pronounced choroiditis has typical chorioretinal foci that are arranged around the optic nerve (Figure 14) [131]. This is also the active driver of vasculitis. The large retinal vessels are predominantly affected. This also explains the sometimes pronounced vitreous involvement. At the same time, the pronounced exudation is also a major etiology for the frequent macular edema, which occurs in around 80% of untreated patients. The HLA-A 29 antigen is considered a classic genotypic characteristic of for BSCR. Occlusive vascular changes are rarely observed in BSCR.

The pathogenesis of the disease is not clear. However, there is much evidence that it has an autoimmunological background. In peripheral blood, increased (IL-21, IL-23, TGF-ß) levels were detected in active BSCR, which indicates a decisive role for the IL-17 pathway [131]. OCT and FA are good options to monitor the activity of the disease such as papillitis, retinal vasculitis and CME (Figure 15 and Figure 16) [132]. In untreated individuals, CME is a frequent finding (30–40%). Vasculitis is to be regarded more as a supportive and less diagnostic criterion for BSCR, but well suited for monitoring the disease and can also prove to be a rarely occurring CNV. More recent technology using UWFFA has shown that peripheral ischemia may present in eyes with chronic BSCR, but it seems not to progress to neovascular complications [133].

Since the pathogenesis originates in the choroid, the detection of typical BSCR lesions with ICGA is more sensitive. Areas of inflammatory, non-perfused choroid can be seen as oval-round lesions in ICGA and often exceed the number of foci that can be seen by fundoscopy (Figure 17). Significant smaller retinal venular calibers have been reported in BSCR eyes compared to matched controls, and the arteriole-to-venule ratio decreased further during follow-up. Whether these changes relate to visual function and treatment outcome still must be proven [134].

In accordance with the assumed autoimmunological genesis of the disease, basic therapeutic agents are applied as cDMARDs and increasingly as bDMARDs. It is still unclear how long therapy should be maintained.

#### 3.2.4. Eales’ Disease

Eales’ disease is also classified as an isolated vasculitis of the eye. It mainly affects younger men and has a different geographic predilection. There is often evidence of an etiology associated with mycobacterium tuberculosis. While this clinical picture was also seen more frequently in Europe in the post-war years (with a relatively high prevalence of tuberculosis), this clinical picture is more likely to be found in India today. Classic clinical changes are found as occlusive vasculitis. The venous vessels are predominantly affected. Vascular nonperfusion and secondary neovascularization are typical clinical changes [135,136,137].

### 3.3. Masquerade Syndromes

The term masquerade covers diseases that cause conditions such as uveitis but do not formally belong to the latter. Uveitis masquerade syndrome (UMS) is an important differential diagnosis especially in elderly patients with uveitis [138]. There are two subtypes of UMS: neoplastic and non-neoplastic [139]. Among neoplastic diseases, those of the lymphoid system should be emphasized. Primary intraocular lymphoma (PIOL) has proven to be the most common etiology of UMS [138,140]. In addition, uveal lymphoid proliferations as well as secondary infiltrations in systemic lymphomas should be mentioned. Generally, these diseases originate from B-cells, but rarely, T-cells are causative. Non-lymphoid, neoplastic etiologies of UMS include choroidal melanoma, metastases, or retinoblastoma. Non-neoplastic forms are, for instance, retinitis pigmentosa, ocular ischemic syndrome, Coats disease, retinal detachment, intraocular foreign body, or posttraumatic [139].

Diagnosis is challenging and often extends over years. Overall, the combination of patient history, slit lamp examination, and multimodality imaging is essential. In any case of suspicion, examinations for staging and imaging of the cranium, should be performed. However, ocular involvement may precede CNS infiltration in 42–92% of cases with PIOL [141]. In PIOL, the IL10/IL6 quotient is increasingly performed in addition to cytology from the vitreous or a retinal biopsy, which is often complicated [142]. Slit lamp examination may reveal changes in the vitreous such as opacities, hemorrhages, or neovascularization. In addition, retinal or choroidal infiltrates and macular edema may be found. In the anterior segment of the eye, conjunctivitis, chemosis, and anterior chamber inflammation with cells, flare, or synechiae may occur.

However, vasculitis has also been described. Case reports speak of diffuse retinal partly occlusive vasculitis with perivascular exudates and retinal whitening in intraocular lymphoma [143,144]. However, the possibility of idiomatic vasculitis and lymphoma activation due to steroid therapy should also be discussed [141]. Say et al. reported a case of vasculitis with diffuse vascular sheathing in a patient with metastatic B-cell lymphoma [145]. The patient also presented with elevated c-ANCA levels. However, ANCA positivity may also accompany lymphoma and does not necessarily underlie systemic vasculitis. Vasculitis has been described in the setting of uveal melanoma as well as ocular metastases. Differentially, an activation of retinal vasculitis in the context of a checkpoint inhibitor therapy must be considered [146,147]. Furthermore, there are case reports of patients with ocular ischemic syndrome or retinitis pigmentosa and retinal partially occlusive vasculitis [148,149].

Therapy requires medication or surgical treatment of the underlying condition. For example, primary intraocular lymphoma will be treated locally with radiation, intravitreal methotrexate, and systemic chemotherapy, depending on the situation [150]

## Figures and Tables

**Figure 1 jcm-11-02525-f001:**
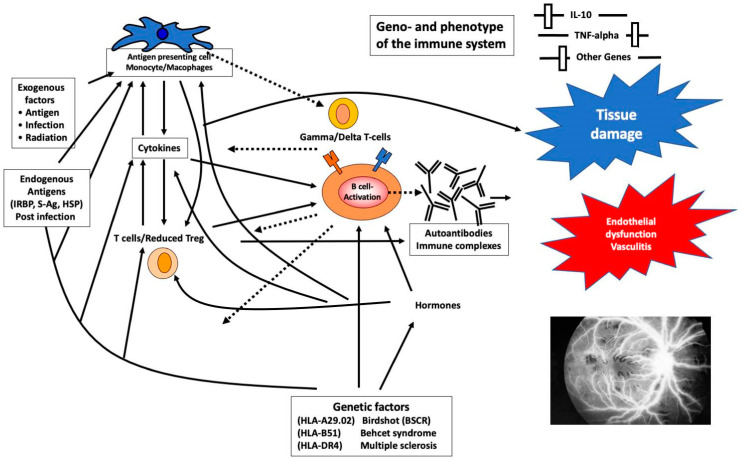
Schematic representation of the pathophysiology and influencing factors in retinal vasculitis. Highly simplified representation of the exogenous and endogenous factors that lead to an immune response in retinal vasculitis. Innate and acquired immunity as well as genetic factors result in vasculitis and subsequent tissue damage. Exogenous factors such as foreign antigens and infectious agents are processed by antigen-presenting cells, which result in B-cell activation. Endogenous antigens such as retinal S antigen can also result in immune stimulation and cytokine release. Both endogenous and exogenous antigen-mediated immune stimulation are controlled by genetic factors. Resultant cytokine release and antigen–antibody complexes result in tissue (endothelial) dysfunction, which is a hallmark of retinal vasculitis. HLA, human leukocyte antigen; HSP, heat shock protein; IL, interleukin; IRBP, interphotoreceptor retinoid binding protein; S-AG, retinal protein S-Ag.

**Figure 2 jcm-11-02525-f002:**
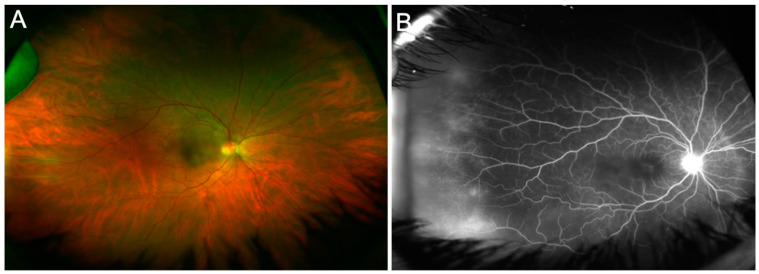
The figure shows a patient (20-year-old Asian Indian male) with tubercular retinal vasculitis. Fundus examination (**A**) reveals optic nerve hyperemia with retinal vascular sheathing and few hemorrhages. Fluorescein angiography (**B**) reveals active optic nerve head inflammation and peripheral retinal vasculitis.

**Figure 3 jcm-11-02525-f003:**
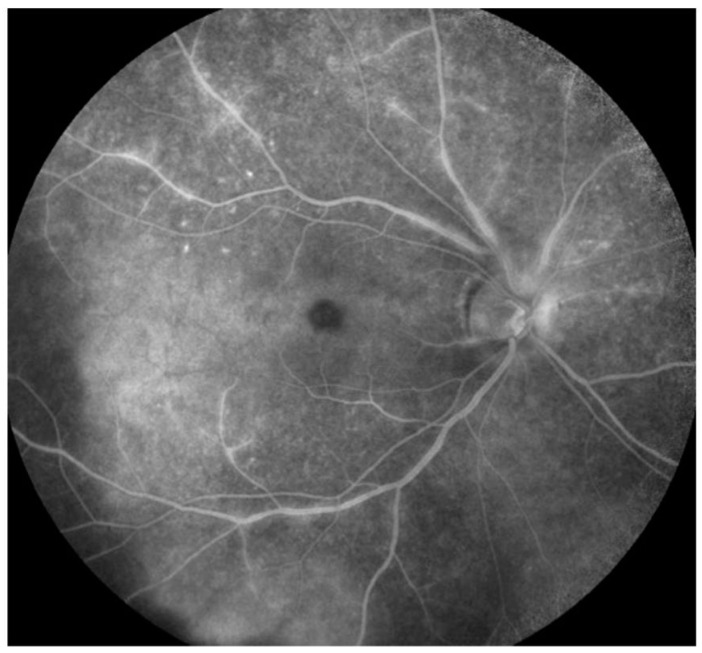
The figure is a fluorescein angiogram (FA) that shows a patient with multifocal retinitis and retinal vasculitis due to ocular syphilis. The FA shows multifocal retinal vasculitis involving the large vessels and multiple areas of hyperfluorescence due to retinitis lesions. There is mild disc hyperfluorescence as well.

**Figure 4 jcm-11-02525-f004:**
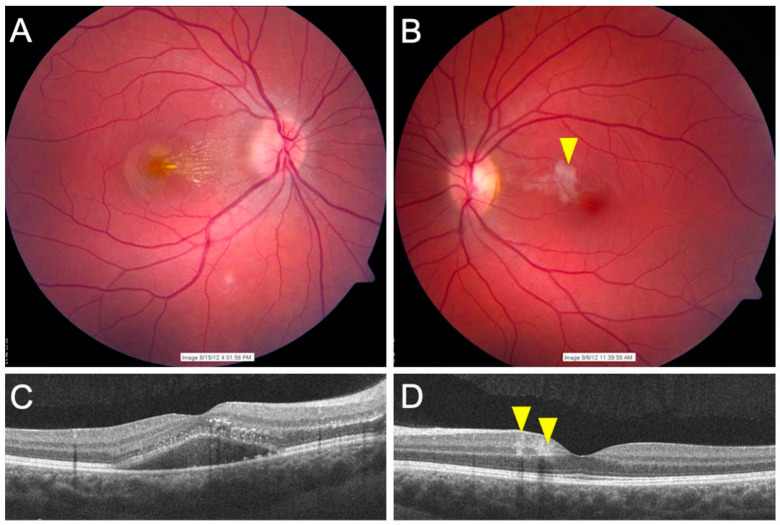
The figure shows a patient with cat-scratch disease (bartonella-related retinitis). Fundus photographs of the right and left eye (**A**,**B**) reveal the presence of neuroretinitis in the right eye and a branch retinal arteriole occlusion in the left eye (yellow arrowheads). The optical coherence tomography of both the eyes (**C**,**D**) does not show macular edema. However, there is significant subretinal fluid in the right eye (**C**). The left eye (**D**) shows hyperreflectivity in the inner retina (yellow arrowhead), suggestive of arteriolar occlusion.

**Figure 5 jcm-11-02525-f005:**
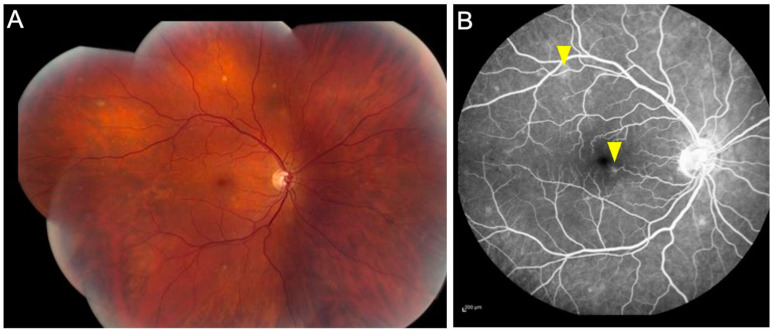
The figure shows a patient diagnosed with Whipple’s disease (**A**). The fluorescein angiography (**B**) shows subtle vascular leakage suggestive of focal vasculitis (yellow arrowheads).

**Figure 6 jcm-11-02525-f006:**
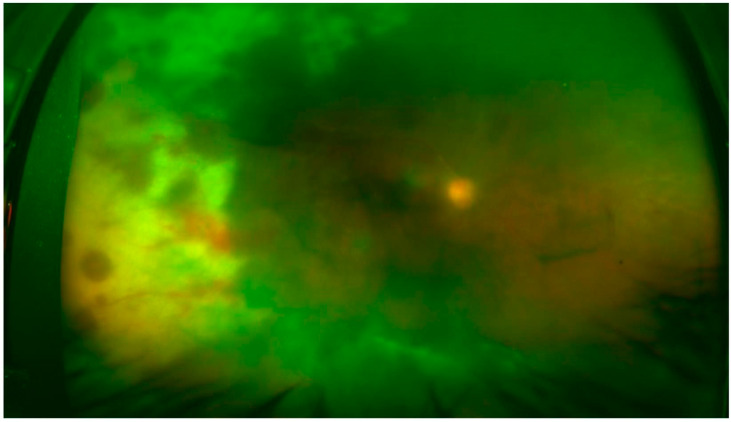
Ultrawide-field imaging of a patient with varicella zoster-related acute retinal necrosis. The fundus photograph reveals extensive peripheral retinal necrosis resulting in whitening and edema, along with retinal arteritis. There is significant media haze due to the presence of vitreous inflammation and cells.

**Figure 7 jcm-11-02525-f007:**
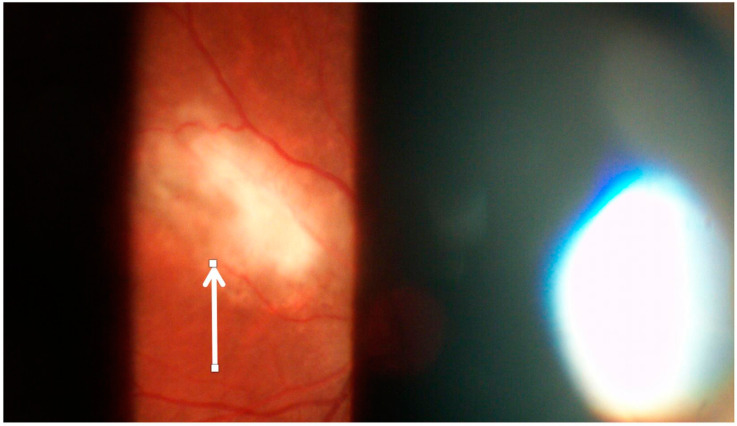
The slit-lamp biomicroscopy fundus photograph using 90 diopter lens shows a necrotizing retinochoroiditis lesion due to toxoplasmosis located nasally to the optic nerve head (white arrow). The active yellow-white lesion is located adjacent to a dull yellow scar, and there is associated retinal vasculitis and overlying vitritis.

**Figure 8 jcm-11-02525-f008:**
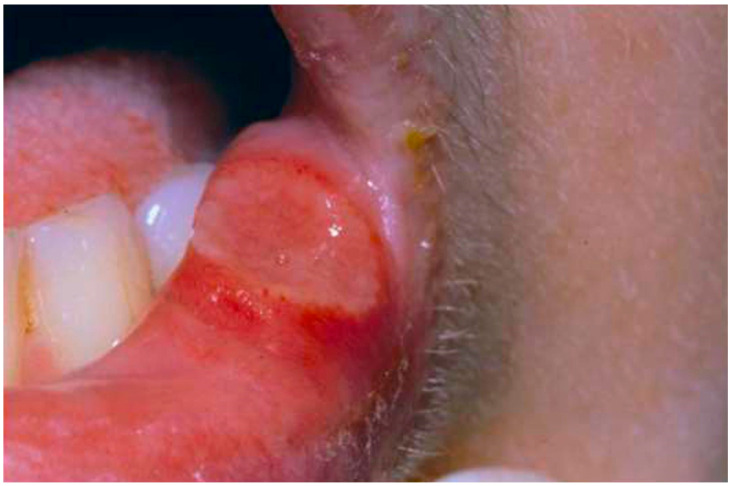
Clinical photograph of 28-year-old male patient with Behcet’s syndrome shows presence of punched out aphthous ulcerations on the lip mucosa.

**Figure 9 jcm-11-02525-f009:**
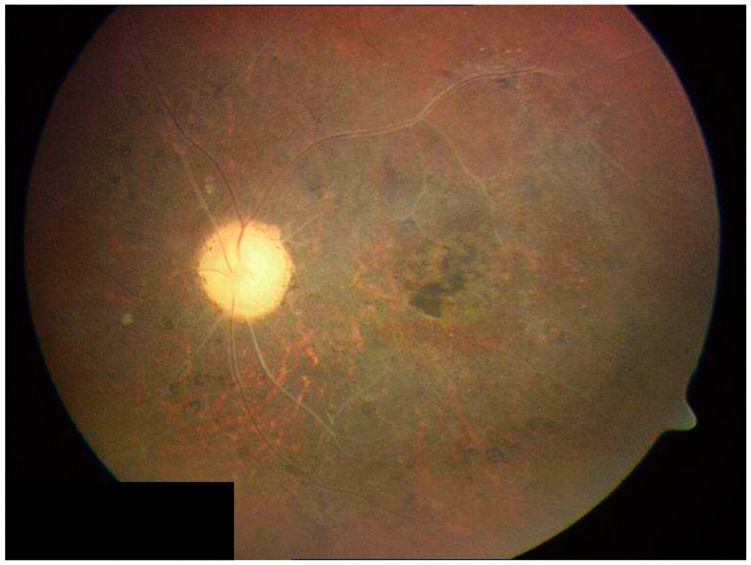
Fundus photograph of a 32-year-old patient with Behcet’s syndrome (only left eye shown) demonstrating occlusive retinal vasculitis leading to severe retinal and optic nerve atrophy. There is severe macular atrophy as well.

**Figure 10 jcm-11-02525-f010:**
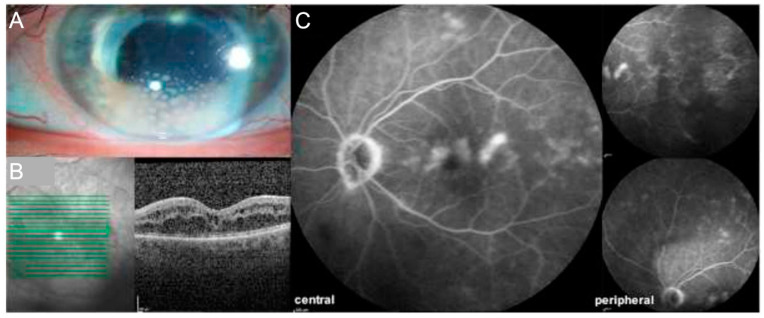
(**A**) 61-year-old woman with panuveitis associated with multiple sclerosis (MS) demonstrating granulomatous keratic precipitates on slit-lamp photography (**A**) and cystoid macular edema on optical coherence tomography (OCT) (**B**). Fluorescein angiography (FA) (**C**) depicts macular edema, optic disc leakage and vascular leakage of central and peripheral vessels. Furthermore, vitreous haze, indicated by a reduced visibility of the fundus best appreciated on peripheral FA and OCT images, is present.

**Figure 11 jcm-11-02525-f011:**
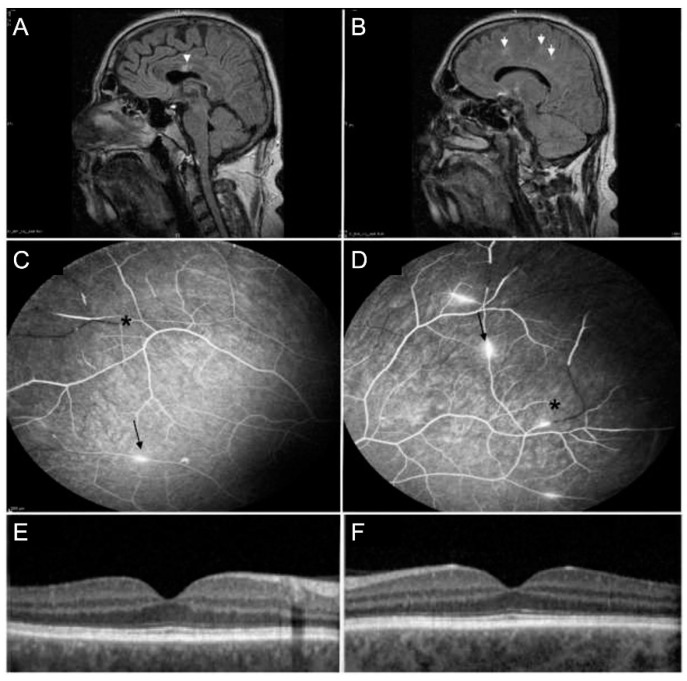
A 39-year-old man with Susac’s syndrome demonstrating hyperintense infarcts in the corpus callosum referred to as “snowball lesions” (white arrows, (**A**)) and hyperintense lesions in the white matter (white arrows, (**B**)) on T2 flair weighted magnetic resonance tomography (MRI) images. Ophthalmologic findings include multiple branch retinal artery occlusions (black asterisk) in the right (**C**) and left eye (**D**). In addition, there are areas of focal arterial wall hyperfluorescence (black arrow) on fluorescein angiography (FA) of both eyes (**C**,**D**). Optical coherence tomography (OCT) of both eyes (right eye: (**E**), left eye: (**F**)) is unremarkable.

**Figure 12 jcm-11-02525-f012:**
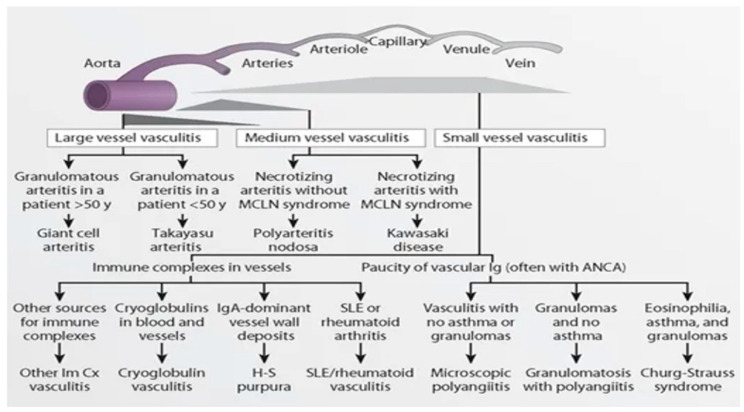
Figure shows the Chapel Hill classification of retinal vasculitis. *(Permission obtained for reprint)*.

**Figure 13 jcm-11-02525-f013:**
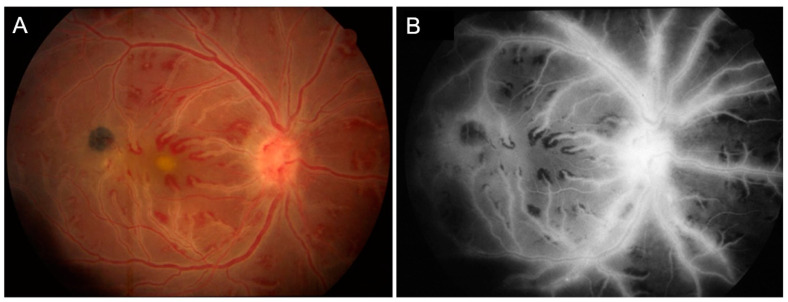
The figure shows fundus photograph of a patient with sarcoid-associated retinal vasculitis with exuberant retinal vessel sheathing, (**A**) suggestive of frosted branch angiitis. Fluorescein angiography (**B**) shows extensive retinal vascular leakage due to the intense inflammatory response.

**Figure 14 jcm-11-02525-f014:**
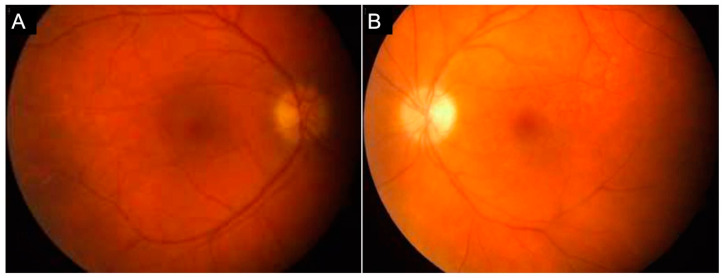
Initial presentation of a 43-year-old female patient with birdshot chorioretinopathy. Fundus photography of the right (**A**) and left eyes (**B**) showing discrete yellowish ovoid choroidal lesions in the peripapillary and midperiphery area.

**Figure 15 jcm-11-02525-f015:**
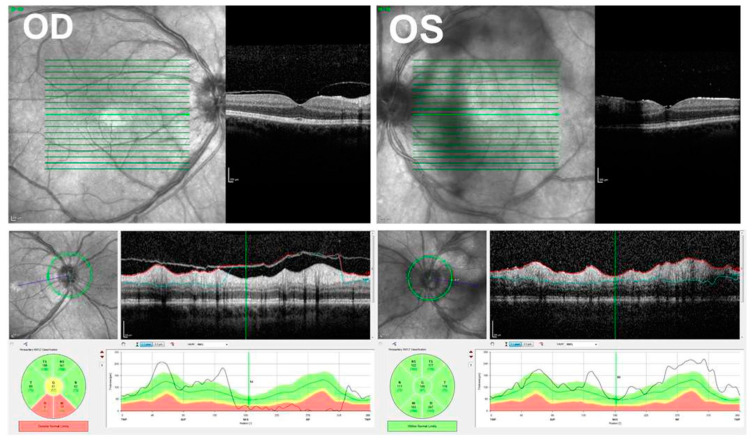
Swept-source OCT of the same patient with birdshot chorioretinopathy shown in Figure 14. Retina: B-scan shows retinal changes. There are intraretinal cysts and hyperreflective foci in outer layers. Optic nerve of the right eye shows reduced nerve fiber layer.

**Figure 16 jcm-11-02525-f016:**
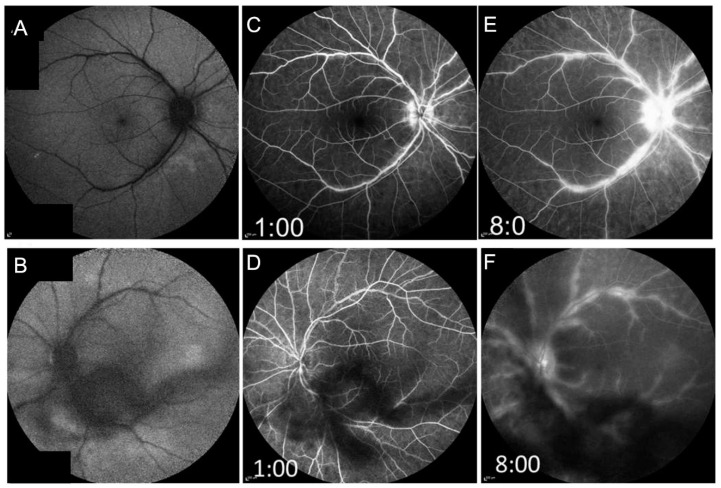
Fundus autofluorescence (FAF) and fluorescein angiography (FA) of the right and left eyes of the same patient with birdshot chorioretinopathy shown in Figure 14 and Figure 15. The FAF is shown in (**A**,**B**) (right and left eyes, respectively). Early phase (**C**,**D**) and late-phase FA (**E**,**F**) of both fundi demonstrate leakage of the optic discs, large vessel leakage, and capillary ferning. Note: floaters are partly impairing visualization.

**Figure 17 jcm-11-02525-f017:**
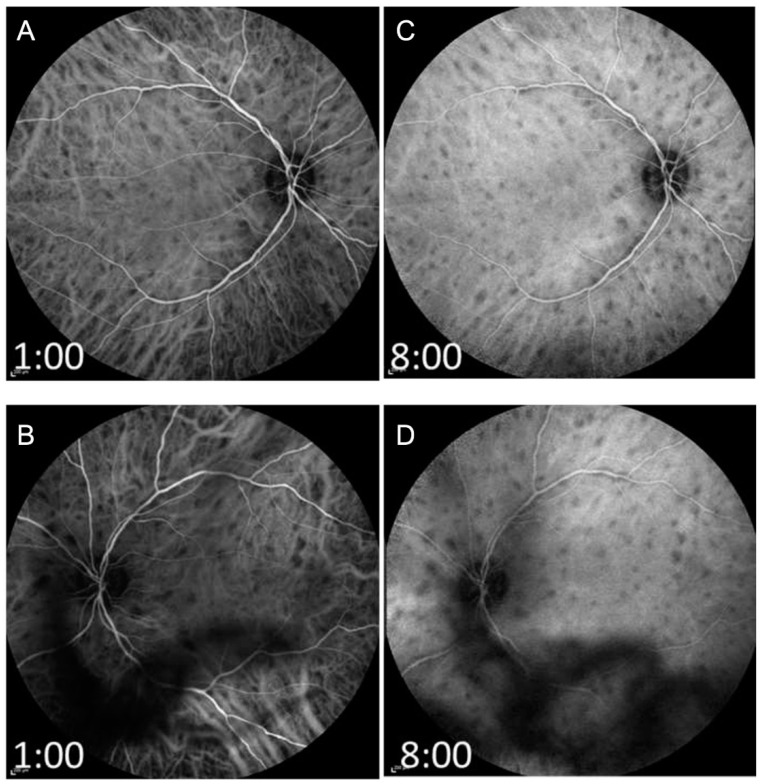
Indocyanine green angiography (ICGA) of the patient with Birdshot chorioretinopathy described in Figure 14, Figure 15 and Figure 16 is shown. The ICGA of both fundi in the early phase (**A**,**B**), and the late phase (**C**,**D**) (right and left eyes, respectively) show numerous evenly distributed and symmetric hypocyanescent spots throughout the fundus of both eyes, suggestive of choroidal inflammation.

**Table 1 jcm-11-02525-t001:** Disorders associated with retinal vasculitis.

Infectious Disorders
*Bacterial disorders* Borreliosis (Borellia burgdorferi) Brucellosis Cat scratch disease (Bartonella henselae) Endophthalmitis Syphilis Tuberculosis Whipple’s disease
*Viral disorders* Acquired immunodeficiency syndrome Acute retinal necrosis Cytomegalovirus Retinitis Chikungunya Dengue fever Hepatitis Human T cell lymphoma virus type 1 Rift Valley fever West Nile Virus
*Parasitic disorders* Mediterranean spotted fever Rickettsia disorders Rocky Mountain spotted fever Toxoplasmosis
**Noninfectious disorders**
*Systemic inflammatory diseases* Behçet’s disease Churg-Strauss syndrome Crohn’s disease Dermatomyositis Granulomatosis with polyangiitis (GPA) HLA-B27-associated uveitis Multiple sclerosis Sarcoidosis Systemic lupus erythematosus Relapsing polychondritis Sjögren’s syndrome Polymyositis Postvaccination Rheumatoid arthritis Susac’s syndrome Takayasu’s disease
*Isolated ocular disorders* Acute multifocal hemorrhagic retinal vasculitis Birdshot chorioretinopathy Frosted branch angiitis (IRVAN) Idiopathic recurrent branch retinal arterial occlusion Intermediate Uveitis
*Masquerade Syndromes* Leukemia Ocular lymphoma (B- T-cell) Paraneoplastic syndromes

## Data Availability

Not applicable.

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
