# Peer review of "A Comprehensive Update on Retinal Vasculitis: Etiologies, Manifestations and Treatments"

_jcm, 2022, doi:10.3390/jcm11092525_

Round 1
Reviewer 1 Report
The text is well reviewed and well structured.
The most important point is figure legends, figure itself, and description in the text are scattered.
Figures needs to be reorganized and reviewed again.
The figures should include the ARN (retinal necrosis), SLE and sarcoidosis.
Figure 1
In the text, ‘In this review we have comprehensively discussed the etiologies, clinical manifestations, and presenta-39 tions of retinal vasculitis (Figure 1).’
There is no detail in figure legend.
Figure 2 OK
Figure 3 Bartonella ?
Figure 4 Wippel? 
Figure 3. The figure shows a patient with cat-scratch disease. Fundus photographs of the right and left eye (A and B) reveal presence of neuroretinitis in the right eye, and a branch retinal arteriole occlusion in the left eye (yellow arrow-heads). The optical coherence tomography of both the eyes (C and D) does not show macular edema. However, left eye (D) shows hyper-reflectivity in the inner retina (yellow arrowhead) suggestive of arteriolar occlusion.
Please add comment, regard to subretinal fluid in figure C (OCT in the right eye).
Figure 5 Behcet’s s
Figure 4. The figure shows a patient diagnosed with Whipple’s disease (A). The fluorescein angiography (B) shows subtle vascular leakage suggestive of focal vasculitis (yellow arrowheads).
Figure 6  Behcet’s s
Figure 5. Photography of patient with Behcet’s syndrome: Aphthous ulcerations on lip mucosa.
Figure 7  
The figure and description do not match.
Figure 8 oral aphtae
(figure 5 in text)
Figure 9
The figure and description do not match.
Figure 10 MS
Figure 7. 61-year-old woman with panuveitis associated with multiple sclerosis (MS) demonstrating granulomatous keratic precipitates and cystoid macular edema on optical coherence tomography (OCT). Fluorescein angiography (FA) depicts macular edema, optic disc leakage and vascular leakage of central and peripheral vessels. Furthermore, vitreous haze, indicated by a reduced visibility of the fundus best appreciated on peripheral FA and OCT images, is present.
(unknown)
Figure 6. Fundus photography of patient with Behcet’s syndrome: left eye with occlusive vasculitis and optic nerve atrophy.
Figure 11 MS
The figure and description do not match.
Figure 12
Figure 9 shows the Chappel Hill classification of retinal vasculitis. (Permission obtained for reprint).
Figure 13 birdshot
Figure 14
The figure and description do not match.
Author Response
Editor-in-Chief
Journal of Clinical Medicine
RE: jcm-1645432, titled A Comprehensive Update on Retinal Vasculitis: Etiologies, Manifestations and Treatments”
Dear Editorial Board:
On behalf all the authors, we thank you and the reviewers for reviewing our manuscript. We appreciate the comments and suggestions and have addressed them as indicated below.
- The text is well reviewed and well structured. The most important point is figure legends, figure itself, and description in the text are scattered. Figures needs to be reorganized and reviewed again. The figures should include the ARN (retinal necrosis), SLE and sarcoidosis.
Response #1: Thank you very much for the detailed review and the constructive comments. We have reorganized the figures as requested. We have added additional figures for the entities mentioned. Figures on ARN and sarcoidosis have been added. We have 17 figures in all now.
- Figure 1
In the text, ‘In this review we have comprehensively discussed the etiologies, clinical manifestations, and presentations of retinal vasculitis (Figure 1).’
There is no detail in figure legend.
Response #2: Thank you very much for the comment. We have modified the figure legend to explain this in more details.
- Figure 2 OK
Response #3: Thank you very much. No changes made.
- Figure 3 Bartonella ?
Response #4: Figure 4 represents cat-scratch disease due to Bartonella. The figure 3 represents syphilis.
- Figure 4 Whipple? 
Response #5: Figure 5 represents Whipple’s disease. The order of the figures has been corrected.
- Figure 3.The figure shows a patient with cat-scratch disease. Fundus photographs of the right and left eye (A and B) reveal presence of neuroretinitis in the right eye, and a branch retinal arteriole occlusion in the left eye (yellow arrow-heads). The optical coherence tomography of both the eyes (C and D) does not show macular edema. However, left eye (D) shows hyper-reflectivity in the inner retina (yellow arrowhead) suggestive of arteriolar occlusion. Please add comment, regard to subretinal fluid in figure C (OCT in the right eye).
Response #6: We have explained the presence of subretinal fluid in the right eye. Thank you.
- Figure 4.The figure shows a patient diagnosed with Whipple’s disease (A). The fluorescein angiography (B) shows subtle vascular leakage suggestive of focal vasculitis (yellow arrowheads).
Response #7: The figure on Whipple’s disease has been relabelled as Figure 5 instead of 4.
- Figure 5 Behcet’s s
Figure 6 Behcet’s s
Response #8: We have relabelled the figures on Behcet’s disease as Figure 8 and 9.
- Figure 7 The figure and description do not match.
Figure 8 oral aphtae
(figure 5 in text)
Figure 9
The figure and description do not match.
Figure 10 MS
Response #9: All the figures have now been correctly labelled throughout the manuscript.
Reviewer 2 Report
This review provides a very well structured overview of the causes and treatment options for retinal vasculitis. It deals with all causes in sufficient depth for a review. It is written clearly and understandably. I have no criticisms of this review.
Author Response
Editor-in-Chief
Journal of Clinical Medicine
RE: jcm-1645432, titled A Comprehensive Update on Retinal Vasculitis: Etiologies, Manifestations and Treatments”
Dear Editorial Board:
On behalf all the authors, we thank you and the reviewers for reviewing our manuscript. We appreciate the comments and suggestions and have addressed them as indicated below.
Reviewer 2:
This review provides a very well structured overview of the causes and treatment options for retinal vasculitis. It deals with all causes in sufficient depth for a review. It is written clearly and understandably. I have no criticisms of this review.
Response: We would like to thank the reviewer for the kind comments and a detailed review of our manuscript. Thank you very much.
Round 2
Reviewer 1 Report
This second version of the paper is a much improvement, the authors are to be commended.
However, there are still a few comments,
Figure numbers have been appropriately corrected but not fully explained.
Instead of just stating FA or OCT, it should be stated as Figure 1A, Figure 1B (as in Figure 3 and Figure 5). The author should revise Figures 10, 11, 13, 14, 15, 16, and 17, along with the text and figure descriptions.
In Fig. 2, macular leakage is not clear in FA, so either the image should be replaced or the comment should be changed.
Author Response
RE: jcm-1645432, titled A Comprehensive Update on Retinal Vasculitis: Etiologies, Manifestations and Treatments”
Dear Editorial Board:
On behalf all the authors, we thank you and the reviewers for reviewing our manuscript. We appreciate the comments and suggestions and have addressed them as indicated below.
- This second version of the paper is a much improvement, the authors are to be commended. However, there are still a few comments, Figure numbers have been appropriately corrected but not fully explained. Instead of just stating FA or OCT, it should be stated as Figure 1A, Figure 1B (as in Figure 3 and Figure 5). The author should revise Figures 10, 11, 13, 14, 15, 16, and 17, along with the text and figure descriptions. In Fig. 2, macular leakage is not clear in FA, so either the image should be replaced or the comment should be change
Response #1: Thank you very much for the comments. We have explained all the figures, and modified the figures 10, 11, 13-17 as desired. The figure 2 has been corrected to remove “macular leakage”.
This manuscript is a resubmission of an earlier submission. The following is a list of the peer review reports and author responses from that submission.